# Transcriptional Regulation of the Synaptic Vesicle Protein Synaptogyrin-3 (*SYNGR3*) Gene: The Effects of NURR1 on Its Expression

**DOI:** 10.3390/ijms23073646

**Published:** 2022-03-26

**Authors:** Lingfei Li, Philip Wing-Lok Ho, Huifang Liu, Shirley Yin-Yu Pang, Eunice Eun-Seo Chang, Zoe Yuen-Kiu Choi, Yasine Malki, Michelle Hiu-Wai Kung, David Boyer Ramsden, Shu-Leong Ho

**Affiliations:** 1Division of Neurology, Department of Medicine, University of Hong Kong, Hong Kong SAR, China; li.lingfei.zj@gmail.com (L.L.); hwl2002@hku.hk (P.W.-L.H.); liuhf@hku.hk (H.L.); pangyys@ha.org.hk (S.Y.-Y.P.); eunseo@connect.hku.hk (E.E.-S.C.); zoecyk@hku.hk (Z.Y.-K.C.); ymalki@connect.ust.hk (Y.M.); mhwkung@hku.hk (M.H.-W.K.); 2Institute of Metabolism and Systems Research, University of Birmingham, Birmingham B15 2TT, UK

**Keywords:** SYNGR3, synaptogyrin, synaptic dysfunction, Parkinson’s disease, gene regulation, promoter analysis, *cis*-regulatory elements, NURR1, transactivator

## Abstract

Synaptogyrin-3 (SYNGR3) is a synaptic vesicular membrane protein. Amongst four homologues (SYNGR1 to 4), SYNGR1 and 3 are especially abundant in the brain. SYNGR3 interacts with the dopamine transporter (DAT) to facilitate dopamine (DA) uptake and synaptic DA turnover in dopaminergic transmission. Perturbed SYNGR3 expression is observed in Parkinson’s disease (PD). The regulatory elements which affect SYNGR3 expression are unknown. Nuclear-receptor-related-1 protein (NURR1) can regulate dopaminergic neuronal differentiation and maintenance via binding to NGFI-B response elements (NBRE). We explored whether NURR1 can regulate SYNGR3 expression using an in silico analysis of the 5′-flanking region of the human SYNGR3 gene, reporter gene activity and an electrophoretic mobility shift assay (EMSA) of potential *cis*-acting sites. In silico analysis of two genomic DNA segments (1870 bp 5′-flanking region and 1870 + 159 bp of first exon) revealed one X Core Promoter Element 1 (XCPE1), two SP1, and three potential non-canonical NBRE response elements (ncNBRE) but no CAAT or TATA box. The longer segment exhibited gene promoter activity in luciferase reporter assays. Site-directed mutagenesis of XCPE1 decreased promoter activity in human neuroblastoma SH-SY5Y (↓43.2%) and human embryonic kidney HEK293 cells (↓39.7%). EMSA demonstrated NURR1 binding to these three ncNBRE. Site-directed mutagenesis of these ncNBRE reduced promoter activity by 11–17% in SH-SY5Y (neuronal) but not in HEK293 (non-neuronal) cells. C-DIM12 (Nurr1 activator) increased SYNGR3 protein expression in SH-SY5Y cells and its promoter activity using a real-time luciferase assay. As perturbed vesicular function is a feature of major neurodegenerative diseases, inducing *SYNGR3* expression by NURR1 activators may be a potential therapeutic target to attenuate synaptic dysfunction in PD.

## 1. Introduction

The synaptogyrin-3 gene (*SYNGR3:*HGNC:11501, NCBI Gene ID: 9143) [1] is a paralogous gene located on the forward strand of chromosome 16p13.3 [2] (position Chromosome 16: 1,989,660–1,994,275 on the Ensembl website, accessed on 1 March 2022) [3] and covers 4615 bp. The Ensembl algorithm indicates that there are six possible transcripts originating from the gene, only one of which can be translated into the SYNGR3 protein [3]. The protein itself is comprised of 239 amino acid residues, and its predicted shape is illustrated in the Uniprot website [4]. SYNGR3 is one of four homologues (SYNGR1 to 4) in the synaptogyrin family of proteins identified in mammals. SYNGR3 shares a common topology of four transmembrane domains with cytoplasmic-exposed N- and C-terminal tails facing the cytoplasmic side of the synaptic vesicular membrane [5,6] (Figure 1).

Although the physiological function of SYNGR3 is not fully understood, SYNGR3 is listed as interacting with seventeen proteins listed on the Pathways Common website [7]. Several of the proteins interacting with SYNGR3 are involved in brain development and neurological diseases, for example Stiff Person Syndrome. SYNGR3 is expressed in synaptic vesicles (SV), where it is located on the vesicular membrane and is involved in SV trafficking [8]. Importantly, though not listed on the Pathways Common website, SYNGR3 has been shown to interact with the dopamine transporter (DAT) and be involved in dopamine (DA) re-uptake and recycling in mouse neuronal MN9D and rat PC12 cells [9]. This study showed that the N-terminal of the SYNGR3 protein is sufficient to interact with DAT [9], whereas the function and interactome of the unstructured C-terminal of SYNGR3 protein are still unclear.

Synaptic vesicles (SV) are important components in presynaptic terminals for the delivery of neurotransmitters to the neural synapse. Perturbation of the expression levels of synaptic vesicle proteins may trigger presynaptic dysfunction in neurological diseases such as Parkinson’s disease (PD) [10], Dementia with Lewy Bodies [11], Alzheimer’s disease (AD) [12,13], and Huntington’s disease [14]. Impairment of SV trafficking causes presynaptic dysfunction and is one of the earliest pathological processes of several neurodegenerative diseases [15]. Decreased SYNGR3 expression has been found in the substantia nigra of a MPTP-induced PD mouse model [16] and in a mouse model treated with 6-hydroxydopamine [17]. In human studies, reduced levels of SYNGR3 expression were found in the brain of PD patients [18]. In a meta-analysis of human brain transcriptome studies, SYNGR3 was found to be a key regulator that is down regulated in PD [19] and in AD patients [20,21]. Furthermore, SYNGR3 was shown to interfere with the association of tau with presynaptic vesicles to alleviate tau-induced defects in vesicle mobility, and to restore neurotransmitter release in a mouse model of AD [22].

The genomic regulatory region of *SYNGR3* and how protein expression is regulated in neurons are largely unknown. Nevertheless, the Pathways Common website lists seven transacting factors (Table 1) which exert control of the gene’s transcription out of 24 listed interactions [7]. To elucidate the regulation of *SYNGR3* in greater detail, we carried out an in silico analysis of the 5′-flanking region of SYNGR3, in which we identified CpG-rich regions and three putative ncNBRE sites (potential NURR1 binding sites). We investigated the actions of these sites using a reporter gene assay, which led to further in silico analysis of the non-coding region of the first exon in which a potential novel promoter was identified. This 5′-flanking region of the human SYNGR3 gene was characterized using reporter gene analyses and site-directed mutagenesis. We then investigated the binding of the NURR1 protein to the putative ncNBRE sites using electrophoretic mobility shift assays (EMSA) and site-directed mutagenesis. Finally, whether a synthetic activator of NURR1 (C-DIM12; 1,1-bis(3′-indolyl)-1-(*p*-chlorophenyl) methane) could affect SYNGR3 expression was explored.

## 2. Results

### 2.1. In Silico Analysis of the 5′-Flanking Region of the Human SYNGR3 Gene

In order to explore the potential regulatory mechanisms controlling SYNGR3 expression, a section of human genomic DNA spanning 2000 bp upstream to 2000 bp downstream of the TIS of the SYNGR3 was examined. Analysis revealed four CpG islands located at −143/+289, +305/+654, +962/+1085, and +1448/+1554 relative to the TIS of the human SYNGR3 gene. These are areas of difficulty in which to initiate PCR reactions and hence were avoided where possible when designing amplification reactions (Figure 2A).

In order to explore the potential regulatory mechanisms controlling SYNGR3 expression, a section of the 5′-flanking region of the human SYNGR3 spanning 2000 bp upstream of the published TIS was analyzed using Transfac and MatInspector. Neither CAAT nor TATA boxes were found within 100 bp upstream of TIS. Based on a comparison with the NBRE consensus sequence (5′-AAAGGTCA-3′), three putative non-canonical (nc) NBRE sites (5′-ACAGGTCA-3′ at −1722/−1717, 5′-AGAGGTCA-3′ at −1312/−1307, 5′-ACAGGTCA-3′ at −849/−844) were found. Each ncNBRE had a single base deviation (bold and underlined) from the canonical NBRE. Other putative transcription response elements, including one Ik-2 (−1646/−1635), two Nkx2s (−1621/−1615, −1473/−1467), one MyoD (−1006/−995), two Sp1 sites (−67/−58, −24/−15), and three TF2Bs (−90/−84, −36/−30, −30/−24) were identified (Figure 2B).

### 2.2. Characterization of the 5′-Flanking Region of the Human SYNGR3 Gene

In order to determine the promoter activity of the 5′-flanking region of the human SYNGR3 gene, a genomic DNA fragment 1870 bp upstream of the published TIS was cloned into a promoter reporter plasmid pGL3-basic (Figure 3A). The resultant plasmid was transfected into SH-SY5Y and HEK293 cells and subjected to luciferase bioluminescence reporter gene assays. The luciferase activity of the 1870 bp construct (pGL3-hSYNGR3–1870/TIS) as determined by the endpoint assay was even lower than the basal activity presented by the promoter-less pGL3-basic empty vector in neuronal SH-SY5Y cells. In non-neuronal HEK293 cells, the pGL3-hSYNGR3–1870/TIS plasmid showed a similar level of basal luciferase activity compared with that of the pGL3-basic empty vector (Figure 3B).

### 2.3. In Silico and Promoter Analyses Showed XCPE1 Decreased SYNGR3 Promoter Activity in SH-SY5Y and HEK293 Cells

The negative effect of the 1870 fragment on promoter activity prompted an additional in silico analysis with MatInspector. Hence, an additional 159 bp (+159) genomic DNA fragment downstream of the TIS (to ATG) was analyzed. This revealed a potential promoter element, XCPE1, (5′-GCGTCCCGCCC-3′) (+138/+148) on the anti-sense strand, which matched the complementary sequence of a typical XCPE1 element (G/A/T-G/C-G-T/C-G-G-G/A-A-G/C-A/C) [23] (Figure 2B). Hence, a new 2029 bp construct (pGL3-hSYNGR3–1870/ATG) was created and the luciferase reporter gene assays were repeated (Figure 4A). Using the continuous real-time luciferase assay, the cumulative luciferase activity of pGL3-hSYNGR3–1870/ATG was found to be significantly higher than that of empty vector controls in both the SH-SY5Y (1.57-fold, *p* < 0.01) and HEK293 cells (2.31-fold, *p* < 0.01) (Figure 4B).

### 2.4. Site-Directed Mutagenesis Confirmed Functional XCPE1 in the 5′-Untranslated Region between TIS (+1) and ATG (+159) of the SYNGR3 Gene

In order to investigate the role of this potential XCPE1 regulatory site in SYNGR3 transcription, site-directed mutagenesis was performed. The mutation exchanged C for T, one pyrimidine for the other pyrimidine—the original sequence, GCGTCCCGCCC changed to the mutated sequence, GCGTCCTGCCC. The real-time luciferase activity of pGL3-hSYNGR3–1870/ATG-XCPE1-Mut in both SH-SY5Y and HEK293 cells is shown in Figure 4. The cumulative luciferase activity of the mutated XCPE1 (pGL3-hSYNGR3–1870/ATG-XCPE1-Mut) showed significantly lower luciferase activity compared with that of pGL3-hSYNGR3-1870/ATG in the SH-SY5Y (↓43.2%, *p* < 0.05) and HEK293 cells (↓39.7%, *p* < 0.05) (Figure 4B).

### 2.5. Site-Directed Mutagenesis Confirmed Functional Putative ncNBRE Sites in the 5′-Flanking Region of the Human SYNGR3 Gene

To determine whether the three putative ncNBRE sites in the 5′-flanking region affected SYNGR3 gene transcription, luciferase reporter plasmids were constructed with a 2029 bp 5′-flanking region containing different mutated ncNBRE sites (pGL3-hSYNGR3–1870/ATG ncNBRE Mut I and II) (Figure 5A). These were transfected separately into both SH-SY5Y and HEK293 cell lines for the endpoint promoter activity assays. Site-directed mutations in ncNBRE site 2 caused a significant reduction in luciferase activity (pGL3-hSYNGR3–1870/ATG ncNBRE Mut I ↓11%, *p* < 0.05) compared with the activity of the construction containing the native ncNBREs in the SH-SY5Y cells (Figure 5B). Such a reduction was not observed in renal HEK293 cells. After all three putative ncNBRE sites were mutated, luciferase activity was reduced by a similar magnitude (pGL3-hSYNGR3–1870/ATG ncNBRE Mut II ↓17%, *p*  <  0.01) when compared with that of pGL3-hSYNGR3–1870/ATG in the SH-SY5Y cells. Interestingly, in the HEK293 cells, there was no reduction in luciferase activity after all three ncNBRE were mutated (Figure 5B).

### 2.6. NURR1 Protein Specifically Bound to Three Putative ncNBRE Sites in the 5′-Flanking Region of the Human SYNGR3 Gene

To determine whether NURR1 binds to the putative ncNBRE sites in the SYNGR3 5′-flanking region, an EMSA assay was performed using nuclear extracts from HeLa cells overexpressing NURR1 and NE-tagged NURR1 (Figure 6A) and three DNA probes (Figure 6B) containing the potential binding sites. The Western blot of the whole cell lysate, nuclear extracts, and cytoplasmic extracts from NURR1 and NE-tagged NURR1 overexpressing HeLa cells showed a band at 72 kDa corresponding to the NURR1 protein and 74 kDa corresponding to the NE-NURR1 protein (Figure 6A), confirming the presence of NURR1 in the lysates for EMSA.

In the EMSA assay, three major DNA–protein complexes were observed designated as Shift 1, Shift 2, and Shift 3 (Figure 6C, lanes 2, 7, and 12). When a two-hundred-fold molar excess of unlabeled SYNGR3-ncNBRE probes were added to the binding reaction, all three shifted bands disappeared for probe 1 and 2 (Figure 6C, lanes 3, 8). However, there a feint band remained in lane 13 for probe 3, possibly due to incomplete competition of the non-labelled probe 3. The specificity of NURR1 binding to the NBRE-like 1 and NBRE-like 2 sites was confirmed with the addition of either anti-NURR1 or anti-NE antibodies, which produced supershifted bands in reactions with their corresponding biotinated-SYNGR3-ncNBRE probes (Figure 6C, lanes 4 and 5, 9, and 10). For an assessment of NBRE-like 3 in lane 14 and 15, only a feint band was observed at around the same size as the supershift bands of probe 1 and 2 that appeared very weak. Moreover, although lanes 9, 10, 14, and 15 appeared to have a feint band above “Shift 3” which potentially have been a supershift, the membrane was overexposed, and yet these suspected bands, if any, still appeared very weak. Moreover, it is noteworthy that smudges were observed in lanes 2, 7, and 12 at the region near the “supershift” bands in lanes 4, 5, 9, and 10, which were merely non-specific binding resulting from the overloading of protein into the wells. More distinctive supershift bands were shown in lanes 4, 5,9, and 10 with a slightly different mobility.

### 2.7. C-DIM12 (NURR1 Activator) Increased SYNGR3 Protein Expression in SH-SY5Y Cells and Its Promoter Activity

Total cell lysates from C-DIM12-treated SH-SY5Y cells and vehicle control cells were probed with anti-SYNGR3 and anti-actin antibodies. After 72 h of treatment with C-DIM12, SDS-PAGE/Western blot analysis showed a strong band at 25 kD, corresponding to the SYNGR3 protein, whereas only a weak band of the same size was detected in cells treated with the drug vehicle (Figure 7A(i)). Quantitative analysis showed that the SYNGR3 level significantly increased (*p* < 0.05) to approximately 17-fold in C-DIM12-treated cells compared with the SYNGR3 level in vehicle-treated cells (Figure 7A(ii)).

To investigate the effects of the NURR1 activator C-DIM12 on SYNGR3 expression in SH-SY5Y cells, a real-time luciferase assay was used to monitor the promoter activity. The cumulative activity was determined as the area under the real-time luciferase activity curve (AUC). In both groups of cells (with or without C-DIM12 treatment), the real time luciferase activities of pGL3-hSYNGR3–1870/ATG began to increase at 12 h and kept increasing until the end of treatment (36 h after treatment). The cumulative luciferase activity of pGL3-hSYNGR3–1870/ATG significantly increased in cells treated with C-DIM12 (1.2-fold higher) compared with that of cells treated with vehicle (*p* < 0.01) (Figure 7B).

## 3. Discussion

In an attempt to elucidate the regulation of SYNGR3 expression, our initial approach was to carry out an in silico investigation of the 4000 bp nucleotide sequence spanning both the 5′-flanking region and 5′-UTR of *SYNGR3*. This has a high GC content and four CpG islands, which made selecting primers for PCR amplification difficult. About 72% of human gene promoters are associated with CpG islands [24], but the percentage of the GC content in the genes in humans does not necessarily correlate with their expression level [25]. SYNGR3 is a putative epigenetically-regulated gene with typical CpG features (defined as a region of ≥200 bp with ≥50% C + G and ≥0.6 CpG observed/CpG expected), which showed a significant 2.3-fold increase in the expression level after chemical-induced demethylation in human breast cancer MCF-7 cells [26]. It is possible that SYNGR3 expression may be regulated under the repressive mechanism of CpG methylation during neuronal development and aging.

Inspection of a 2000 bp section of the 5′-flanking region showed that it did not contain either a typical TATA or a CAAT box but contained a number of other elements, for example two Sp1 sites. To determine whether the putative *cis*-acting elements we identified were functional, we determined the promoter activity of the 1870-bp genomic DNA fragment upstream of the TIS of SYNGR3 gene. To our surprise, no promoter activity of this segment was observed. This lack of promoter activity indicated that the core elements responsible for basal transcription were not present but might have been located in another region of the gene. This speculation is supported by a previous study which mapped crucial promoter elements responsible for the transcriptional activity to the 5′-untranslated region (5′-UTR) of the human apolipoprotein gene, but not to the usual location upstream of the TIS [27]. Therefore, we examined the effects on transcription with an additional 159-bp 5′-UTR. In contrast to the previous segment, the 2029-bp DNA fragment, which included the 5′-UTR, drove reporter expression in both cell lines, indicating that the 159-bp segment between the TIS and start codon is essential for basal promoter activity. This small section contained a potential XCPE1 site which we hypothesized was the functional unit driving promoter activity.

A substantial fraction of gene promoters that lack a TATA box contain other core promoter elements, such as XCPE1 that function independently of a TATA-box [28]. The SYNGR3 gene has no TATA box and has a high GC content around TIS, which is consistent with the characteristics of genes under XCPE1 regulation. XCPE1 has a 10-bp consensus sequence of (G/A/T)-(G/C)-G-(T/C)-G-G-(G/A)-A-(G/C)-(A/C) which is found in the core promoter regions of approximately 1% of human genes, particularly in TATA-less genes [23,29]. XCPE1 drives transcription cooperatively with some activators, such as the TATA-binding protein and the whole transcription factor II D complex [23]. In both neuronal and renal cell lines, we showed that the mutation of XCPE1 significantly reduced but did not abolish SYNGR3 promoter activity. It has been reported that XCPE1 exhibits little activity by itself, but works in conjunction with sequence-specific activators, such as Sp1 [23]. We identified two Sp1 sites (−67/−58 and −24/−15 to TIS) in the 5′-flanking region of SYNGR3. Thus, it appears that XCPE1 may be working along with these two Sp1 sites to direct transcription initiation.

Three putative ncNBRE sites were identified in the 5′-flanking region of *SYNGR3,* which may bind NURR1. NURR1 is a member of the nuclear receptor superfamily and is mainly located in the nucleus [30]. We obtained two independent nuclear extracts from cells overexpressing either NURR1 (non-tagged) or NE-NURR1 (NE-tagged) for the ncNBRE binding assay. The EMSA showed that both NURR1 and NE-NURR1 proteins bound to the biotin-labeled probes containing the three putative ncNBRE elements (Figure 6C, lanes 2, 7, and 12). Three major shifted bands (Shifts 1–3) were observed, indicating there were three different DNA–protein complexes resulting from the binding reaction. Regarding the supershift assay, one might argue that in supershift lanes, the shift bands should be lighter than those in lanes without antibodies. However, there were no detectable difference between the two in all three probes for Shift 1 and 2, and only a slight difference for Shift 3 of probe 2 and 3. This observation is not surprising because of protein overloading in the EMSA reactions which led to the intensities of the bands with and without antibodies appeared to be saturated under visual examination. To gain good evidence for a supershift band, the membrane has to be overexposed with regard to the gel shift band. Hence the gel-shift bands may appear to have the same intensity.

A number of studies have reported an interaction of NURR1 with different transcription factors and/or coactivators. For instance, NURR1 interacts with the retinoid X receptor (RXR) to form a hetero dimer as a potential target in the treatment of neurodegenerative diseases [31]. Prostaglandin E1 (PGE1) and its dehydrated metabolite, PGA1 have been shown to exhibit neuroprotective effects in a NURR1-dependent manner via an enhancement of the expression of NURR1 target genes in mouse dopaminergic neurons [32]. Moreover, NURR1 has been shown to interact with NF-kappa B to suppress the inflammatory response in microglia [33]. NURR1 can be activated by numerous compounds, although originally it was thought to be a constitutive transactivating agent [34]. NURR1 binds to the NBRE element as a monomer to activate transcription constitutively [35]. Thus, it is likely that the ncNBRE sites bound to NURR1 as a monomer. Among the three shifted bands, the bands of Shift 3 which moved the furthest and, therefore, were the smallest of the three, were most likely generated by monomeric NURR1 binding. The two larger complexes may be generated by the binding of NURR1 together with either its heterodimer partner, RXR [36], and/or ancillary proteins such as co-inhibitors or co-activators. When either anti-NURR1 or anti-NE antibodies were incubated with the reaction mixes, at least one major supershift band was seen, indicating that NURR1 and NE-NURR1 were present in at least one of the shifted bands. The fact that only one major supershift band was seen may be due to the large, high affinity IgG molecule blocking the binding of any other protein to the NURR1. Alternatively, NURR1 itself or the epitope to which the antibody binds may be inside the complexes forming the larger shift bands, negating binding by anti-NURR1 or anti-NE antibodies. NURR1 and NE-NURR1 binding to NBRE-like 3 appeared weaker than the binding to NBRE-like 1 and 2 (Figure 6), suggesting that NBRE-like 3 is a weak NURR1-binding site, if any. Mutation of the three putative ncNBRE sites caused a significant reduction in promoter activity when compared with that of its native promoter, indicating that these binding sites are involved in regulating *SYNGR3* transcription activity. Interestingly, such a reduction was only observed in SH-SY5Y neuronal cells but not in the non-neuronal HEK293 cells. Such a difference is probably due to neuron-specific factors not present in renal cells. Furthermore, when all three putative ncNBRE sites were mutated, the promoter activity decreased by 17% when compared with the native promoter, which was of a similar magnitude to the 11% reduction when only ncNBRE2 was mutated (Figure 5). This suggests that ncNBRE2 contributed the most among the three sites to the transcriptional activity of the 2029-bp segment.

NURR1 is crucial for the development of neuronal stem cells and the survival of mature dopaminergic neurons. Its role has been described with regards to the etiology of PD as early as 2001 [37]. NURR1 forms a heterodimer with the 9-cis retinoic acid receptors, although it is thought to not have a natural ligand unlike many hormonal members of the nuclear receptor superfamily, and similar to them, corepressor and activator proteins are not required to regulate the actions of NURR1. More recently, many of the targets of NURR1 have been identified, revealing that approximately 40 direct target genes of NURR1 and the expression of genes related to synapse formation and neuronal cell migration correlated tightly with NURR1 expression [38]. Another aspect of NURR1 action involves its interaction with NF-kappa B and the suppression of the neuro-inflammation in PD. Upregulation of NURR1 and NF-kappa B reduced neuro-inflammation [39]. Currently a variety of treatments to arrest or slow the progression of PD are being explored. One such example is the replenishment of gangliosides. Intranasal injection of gangliosides GM1 into an α-synuclein mutant mouse decreased α-synuclein levels and increased dopaminergic neuronal survival and NURR1 expression [40]. Deficiency in NURR1 signaling is evident in autopsied PD midbrains and in the peripheral lymphocytes of patients with parkinsonian disorders [41,42]. Involvement of NURR1 in the development and progression of PD makes this nuclear factor together with its downstream regulatory proteins (e.g., SYNGR3) potential targets for therapeutic intervention.

Although the gene mutation of *SYNGR3* has not been reported in human neurodegenerative diseases, post-translational modification which causes alteration in SYNGR3 protein levels has been shown to be associated with various cancers in humans. For example, gene expression profiling revealed that the differential expression of SYNGR3 is a novel immunohistochemical marker which discriminates two pathologic entities in renal cancers [43]. Relative to HPV^−^ head and neck cancers (HNC), HPV^+^ HNC and cervical cancer showed a significantly increased expression of a number of genes, including *SYNGR3* [44]. The differentially expressed mRNAs in the normal cervical epithelium and primary tumors were detected by an mRNA microarray assay which revealed *SYNGR3* as one of the ten most overexpressed potential diagnostic and prognostic biomarkers in cervical cancers compared to the normal cervical epithelium [45].

In this study, we showed that treatment with a NURR1 activator, C-DIM12, in SH-SY5Y cells increased SYNGR3 protein expression. C-DIM12 has been shown to transactivate NURR1 in neuron-like PC12 cells [46] and various cancer cells [47,48]. This compound promotes the NURR1-dependent transrepression of NF-κB-regulated genes in BV-2 microglia which mediate neuroinflammation [49]. The spontaneous pacemaker activity of nigrostriatal DA neurons triggers tonic dopamine release to regulate voluntary movement [50,51], which requires continuous and rapid DA turnover. Any disturbance in these homeostatic processes can result in cytosolic DA accumulation and a greater energy burden, oxidative stress, and synaptic dysfunction in dopaminergic nigrostriatal neurons resulting in PD [52]. The SYNGR3 level has been shown to be reduced in the brains of PD patients [18], implicating a potential pathogenic role to cause synaptic dysfunction. The reasons why SYNGR3 expression is reduced in various human neurological disorders is yet unclear. Although it may be associated with the differential epigenetic regulation of gene promoter activity of SYNGR3 [26], our results indicate that NURR1 transactivators may be helpful to maintain dopaminergic synaptic function via SYNGR3, thus presenting a potential therapeutic target to preserve neuronal function in PD (Figure 8). Synaptic dysfunction exists as an early feature of PD based on neuroimaging studies [53] and in its experimental models [54] which are potentially amenable to intervention [55].

## 4. Conclusions

In conclusion, we demonstrated that the transcriptional activation of the *SYNGR3* gene involves a 2 kb proximal promoter region and 159-bp 5′-UTR which contains a core promoter element specific to most TATA-less gene promoters, XCPE1. The regulatory elements which affect *SYNGR3* expression have not been previously reported. We demonstrated the promoter activity of three putative ncNBREs and the specificity of NURR1 binding to these sites, which likely contributes to SYNGR3 expression in vivo. The presence of these functional ncNBREs that can bind to and respond to NURR1 activation support an important link between SYNGR3 and NURR1. The latter is critical to the maintenance of dopaminergic neuronal health and function, without which dopamine neurons would not develop normally. Treatment of SH-SY5Y cells with the NURR1 activator, C-DIM12, significantly increased SYNGR3 protein expression. Given that perturbed vesicular function is a feature of major neurodegenerative diseases, inducing *SYNGR3* expression by NURR1 activators may be a potential therapeutic target to attenuate synaptic dysfunction in PD. 

## 5. Materials and Methods

### 5.1. Cell Cultures

Human SH-SY5Y cells were cultured in Dulbecco modified Eagle medium-F12 (DMEM-F12) (Invitrogen, Carlsbad, CA, USA) supplemented with 10% fetal bovine serum (FBS) (Invitrogen, Carlsbad, CA, USA) and 100 μg/mL penicillin–streptomycin (Invitrogen, Carlsbad, CA, USA). HEK293 cells and on occasion Human HeLa were cultured in Roswell Park Memorial Institute (RPMI) 1640 medium (Invitrogen, Carlsbad, CA, USA) supplemented with 10% FBS and 100 μg/mL penicillin–streptomycin. All cells were incubated at 37 °C in a humidified 5% CO_2_ atmosphere.

### 5.2. In Silico Analysis of SYNGR3 5′-Flanking Sequence

The presence of CpG islands within a 4000 bp section of human genomic DNA spanning from 2000 bp upstream of the transcription initiation site to 2000 bp downstream of the site were identified using MethPrimer (http://www.urogene.org/cgi-bin/methprimer/methprimer.cgi; accessed on 1 March 2019) [56]. Potential cis-acting response elements in 2000 bp of the 5′-flanking region of *SYNGR3* were investigated using TFsearch [57] and MatInspector [58] programs.

### 5.3. Human SYNGR3 Promoter–Luciferase Fusion Constructs for Reporter Gene Assays

To generate human SYNGR3 promoter–luciferase reporter gene constructs, human genomic DNA was used as the initial template for the PCR amplification of a 2094 bp fragment (1870 bp upstream to 224 bp downstream of the TIS). The resultant fragment was cloned into a pCR4-TOPO vector (Invitrogen, Carlsbad, CA, USA). The insert was sequenced to confirm the veracity of the amplification. The pCR4-TOPO-hSYNGR3–1870/+224 was used as a template to amplify two SYNGR3 5′-flanking fragments of 1870 bp and 2029 bp; the first coding from the TIS to −1870 bp and the second from the translation start codon (ATG; at +159 bp) to −1870. Two PCR products (*NheI*-1870/TIS-*XhoI* and *NheI*-1870/ATG-*NcoI*) were generated with restriction sites at both ends. After digestion with the relevant restriction enzymes (*NheI* and *XhoI*, *NheI*, and *NcoI*; New England Biolabs, Ipswich, MA, USA), the two DNA fragments were ligated into a promoter-less luciferase expression vector (pGL3-basic; Promega Corporation, Madison, WI, USA) to yield two plasmids, pGL3-hSYNGR3–1870/TIS and pGL3-hSYNGR3–1870/ATG. Primers for the PCRs (Table 2) and DNA sequencing were carried out by Tech Dragon Ltd. (Hong Kong).

### 5.4. Human SYNGR3 Gene Promoter Activity Assay

Promoter activity was assessed by assay of the levels of luciferase activity following expression of the SYNGR3 promoter–luciferase fusion constructs in human SH-SY5Y cells. Initial experiments were carried out using the pGL3-hSYNGR3–1870/TIS construct, and later the pGL3-hSYNGR3–1870/ATG construct. The pRL-TK vector (Promega Corporation, Madison, WI, USA), which encodes Renilla luciferase, was co-transfected and used as an internal control for transfection efficiency. Cells were transfected at 70% confluence in 24-well plates using plasmid DNA (μg) to Lipofectamine2000 (μL) ratio of 1:3.

Two types of luciferase assay were used to assess gene promoter activity: an endpoint assay and a continuous real-time assay. For the endpoint assay, promoter activity was assessed using the Dual-Luciferase Reporter (DLR™) Assay System (Promega Corporation, Madison, WI, USA). The luciferase activity of each construct was measured in triplicate in at least three independent experiments. For the real-time luciferase assay, cells were seeded in a 35-mm culture dish 24 h before the transfection of the testing constructs. The culture medium was replaced with a fresh medium containing 0.5 mM D-luciferin (Invitrogen, Carlsbad, CA, USA) 4 h after transfection. The resultant cultures were maintained at normal culture conditions in a real-time luminometer (AB-2550 Kronos Dio, Atto, Tokyo, Japan). Bioluminescence emitted from the whole culture was measured and integrated at 1 min intervals for 36 h.

### 5.5. Site-Directed Mutagenesis of Three Putative ncNBRE Sites in the SYNGR3 Promoter Region

Site-directed mutagenesis was performed using a QuickChange Lighting site-directed mutagenesis kit (Agilent Technologies, Santa Clara, CA, USA) as described above. pGL3-hSYNGR3–1870/ATG-ncNBRE-Mut I (one ncNBRE site was mutated) and pGL3-hSYNGR3–1870/ATG-ncNBRE-Mut II (all three ncNBRE sites were mutated) were investigated. One base was changed in each mutated sequence. Primers for the site-directed mutagenesis are listed in Table 3.

### 5.6. Construction of the Expression Plasmids Encoding NURR1 (pcDNA3.1(+)-NURR1) and NE-NURR1 (pcDNA3.1(+)-NE-NURR1) Proteins

A small polypetide sequence (NE) was attached to the N-terminal of NURR1 to aid immunological detection of the expressed protein in the EMSA experiments. NE is an 18-amino-acid patented epitope tag [55,59] and is available at (Versitech Ltd., Hong Kong; http://www.versitech.hku.hk/reagents/ne/; accessed on 1 March 2022).

Human NURR1 cDNA (pcDNA3.1(+)-NURR1) and NURR1 cDNA conjugated with NE tag (pcDNA3.1(+)-NE-NURR1) were cloned into pcDNA3.1(+) using *HindIII* and *BamH1* restriction sites (New England Biolabs, MA, USA). The primers used are listed in Table 1 (NE-tag—in bold type and underlined). The PCR product was purified using agarose gel electrophoresis and was digested and ligated with the pcDNA3.1 (+) vector. Ligated circular plasmids were transformed into One Shot^®^ OmniMAX™ 2 T1R Chemically Competent Escherichia coli (Invitrogen, Carlsbad, CA, USA). Positive clones were determined by restriction analysis and confirmed by sequencing. Candidate positive colonies were screened by the small-scale preparation of plasmid DNA and endonuclease digestion. DNA sequencing was used to confirm the orientation of the ligated product.

### 5.7. Overexpression of NURR1 and NE-NURR1 in HeLa Cells

The amount of endogenous NURR1 protein in HeLa cells was not sufficient to allow the sensible detection of DNA probe binding in the EMSA assay. To obtain sufficient NURR1 protein to perform the EMSA assay, NURR1 and NE-NURR1 were overexpressed in HeLa cells. Briefly, HeLa cells were seeded at 40–50% confluence and cultured in 6-well plates with a complete RPMI 1640 medium supplemented with 10% FBS and 1% penicillin and streptomycin. Expression plasmids encoding for native NURR1, NE-tagged NURR1, or the empty plasmids was transfected into cells using Lipofectamine2000™. After 48 h, whole cell lysates, nuclear extracts, and cytoplasmic extracts from each of those transfected cells were collected and confirmed by SDS-PAGE/Western blots for NURR1 overexpression. Nuclear and cytoplasmic extracts were prepared using the NucBuster™ protein extraction kit (EMD Biosciences, Darmstadt, Germany). Equal amounts of protein (20 μg) were electrophoresed on 10% (acrylamide/bisacrylamide 37.5:1; Bio-Rad™) SDS-PAGE and transferred onto a PVDF membrane. Resulting blots were blocked with 5% non-fat skimmed milk and probed with anti-NURR1 (1:1000, Santa Cruz Biotechnology, Santa Cruz, CA, USA) and anti-NE antibodies (1:1000) [55,59]. Blots were incubated with horseradish peroxidase-conjugated secondary antibodies (1:5000) followed by ECL substrate detection. Immunoblots were quantified by computerized scanning densitometry.

### 5.8. Treatment of SH-SY5Y Cells with C-DIM12 (NURR1 Activator)

SH-SY5Y cells were treated with a NURR1 activator, C-DIM12 (Bio-Techne Hong Kong Ltd., Hong Kong) to induce NURR1 expression/activation. Cells were seeded at 40–50% confluency in 24-well plates 24 h before treatment. The cells were then exposed to 10 μM C-DIM12 or vehicle (0.01% DMSO) for 72 h. Total protein lysates from treated cells were extracted and quantified. Equal amounts of protein (30 μg) were electrophoresed in 10% (acrylamide/bisacrylamide 37.5:1; Bio-Rad™) SDS-PAGE gels and then transferred onto a PVDF membrane. Resulting blots were blocked with 5% non-fat skimmed milk and probed with anti-SYNGR3 (1:3000), anti-NURR1 (1:1000) or anti-β-actin (1:1000) (Santa Cruz Biotechnology, Santa Cruz, CA, USA) antibodies. Blots were incubated with horseradish peroxidase-conjugated secondary antibodies (1:5000) followed by ECL substrate detection. Immunoblots were quantified by computerized scanning densitometry.

SH-SY5Y cells were transiently transfected with the pGL3-hSYNGR3–1870/ATG construct containing three NBRE-like elements. Real time luciferase activity was measured from 12 h to 36 h following treatment with vehicle and C-DIM12 (10 μM) in SH-SY5Y cells as described above.

### 5.9. Western Blots

Cells were lysed in an ice-cooled 1× RIPA lysis buffer with 0.1% SDS (Cell Signaling Technology; Danvers, MA, USA) with a protease inhibitor cocktail (Roche; Hong Kong, China). The total cell lysates were incubated on ice for 15 min and centrifuged at 4 °C for 15 min at 12,000× *g*. For the extraction of nuclear extracts and cytoplasmic extracts for EMSA, a commercial nuclear extraction kit (Millipore; Hong Kong, China) was used according to the manufacturer’s protocol. The protein concentration of lysates was determined by the Bradford assay (ThermoFisher™ Scientific; Waltham, MA USA). The lysate solution was boiled for 5 min at 80 °C in a 1× denaturing sample buffer (ThermoFisher™ Scientific; Waltham, MA USA). Samples containing each amount of protein were placed in the wells of a 10% polyacrylamide gel (375 mM Tris, 10% Acrylamide/Bis, 0.1% SDS, 0.05% APS and 0.15% TEMED) and electrophoresed in a Tris-Glycine SDS running buffer (25 mM Tris, 190 Mm glycine, and 0.1% SDS; pH 8.3) at 80 V for 30 min followed by 100 V for 90 min. Separated proteins were transferred onto a nitrocellulose membrane by electrophoresis in an ice-cooled Tris-Glycine transfer buffer (25 mM Tris, 190 Mm glycine, and 15% methanol; pH 8.3) at 100 V for 2 h. The membrane was blocked with 5% BSA in TBS and probed with antibodies against SYNGR3 (1:3000, Santa-Cruz Biotechnology; #sc-271046, 26 kD), or actin (1:3000, Santa-Cruz Biotechnology #sc-1615, 43 kD). For chemiluminescence detection, blots were incubated with HRP-conjugated secondary antibodies (DAKO) followed by ECL substrate (ThermoFisher™ Scientific; Waltham, MA USA). Immunoblots were scanned and the scanned images were analyzed by ImageJ software.

### 5.10. Statistical Analyses

All our experiments were completed in at least three independent trials in which each trial was performed in triplicate. Standard error of the mean (S.E.M.) quantifies uncertainty in the estimates of the mean among trials [60]. Therefore, we expressed our data in mean ± S.E.M to express the certainty of calculated means among independent trials. The statistical difference between two independent groups was either assessed by a one-way ANOVA followed by the post hoc Dunnett’s test or direct comparison using the unpaired Student’s *t*-test in GraphPad™ PRISM software ver. 9 (GraphPad Inc., San Diego, CA, USA). Differences were considered significant at a level of *p* < 0.05.

## Figures and Tables

**Figure 1 ijms-23-03646-f001:**
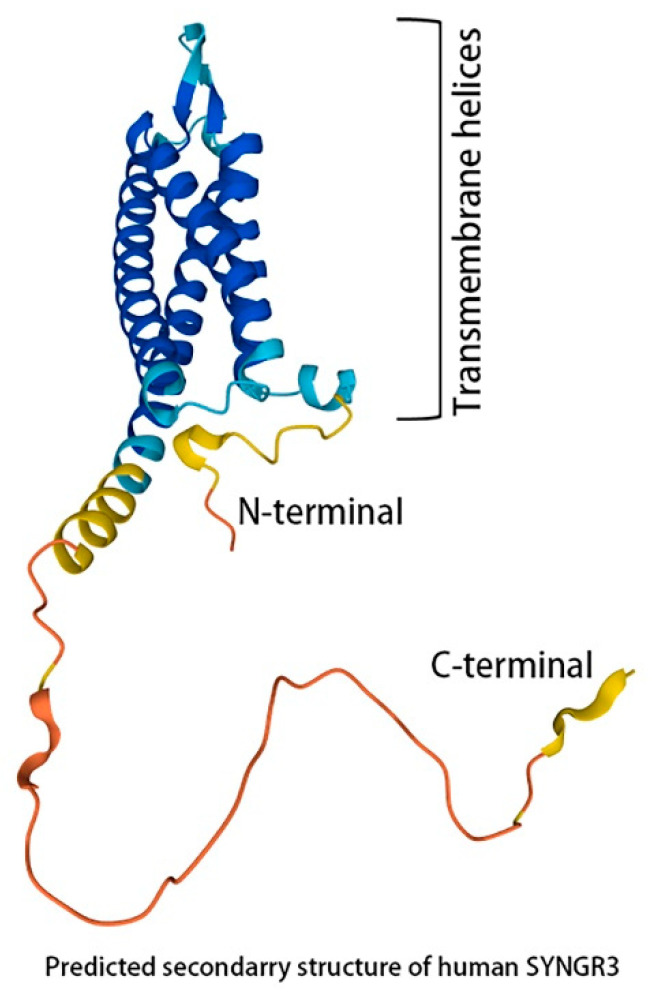
Predicted structure of the human SYNGR3 protein revealed free hanging N- and C-terminal in the cytosol (image source from Uniprot—a freely accessible database of protein sequence and functional information, with modifications).

**Figure 2 ijms-23-03646-f002:**
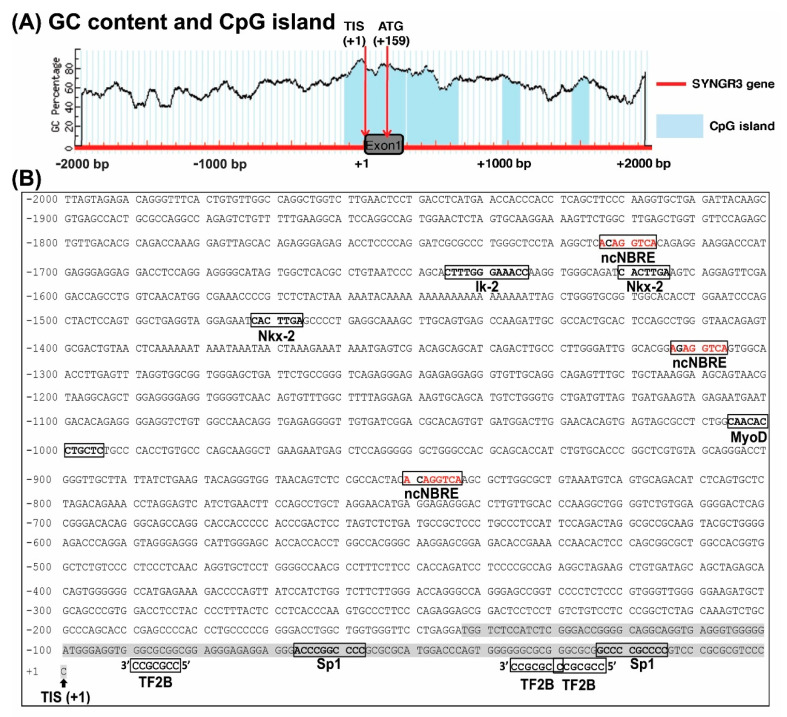
Characterization of the 2000 bp 5′-flanking region of the human SYNGR3 gene. (**A**) Analysis using the MethPrimer program (http://www.urogene.org/methprimer/; accessed on 1 March 2019) revealed four CpG islands located at −143/+289, +305/+654, +962/+1085, and +1448/+1554 relative to the published TIS of human SYNGR3. Blue areas indicate potential CpG islands (island size > 100 bp, GC% > 50%, observed-to-expected CpG ratio > 0.6). Red arrows indicate TIS and ATG sites. (**B**) Genomic DNA sequence and putative transcription factor binding sites in the vicinity of 2000 bp upstream to 200 bp downstream of TIS of the SYNGR3 gene.

**Figure 3 ijms-23-03646-f003:**
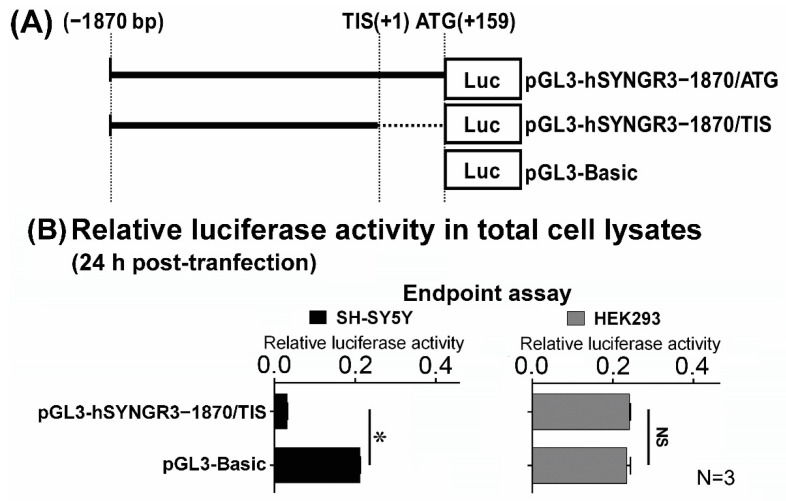
Promoter activity of −1870 bp to the ATG of the SYNGR3 gene as measured by luciferase reporter assays. (**A**) Schematic diagram of the pGL3-Basic and pGL3-hSYNGR3–1870/TIS constructs. (**B**) End point (24 h post-transfection) luciferase activity of pGL3-Basic and pGL3-hSYNGR3–1870/TIS constructs in SH-SY5Y and HEK293 cell lysates. * *p* < 0.05 represents the statistical significance between the two designated groups by the unpaired Student’s *t*-test.

**Figure 4 ijms-23-03646-f004:**
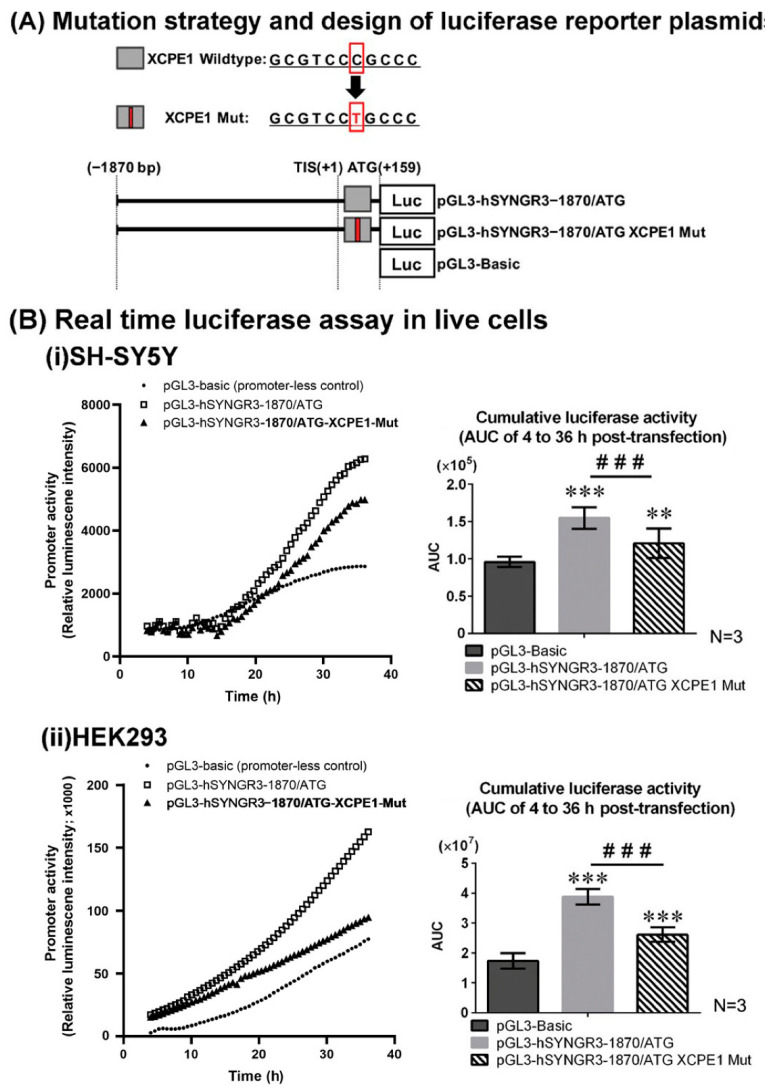
Effects of wildtype and mutated XCPE1 on SYNGR3 promoter activity monitored by a real-time luciferase assay in neuronal SH-SY5Y and non-neuronal HEK293 cells. (**A**) Construction of luciferase-plasmids with XCPE1 site-directed mutation. (**B**) (i) Real-time luciferase assay comparing wildtype and mutant XCPE1 in SH-SY5Y cells; and (ii) in HEK293 cells. Data are expressed as mean ± SEM of three replicates. **, *p* < 0.01, *** *p* < 0.001 compared with control (pGL3-Basic) construct. ###, *p* < 0.001, compared between the two designated groups.

**Figure 5 ijms-23-03646-f005:**
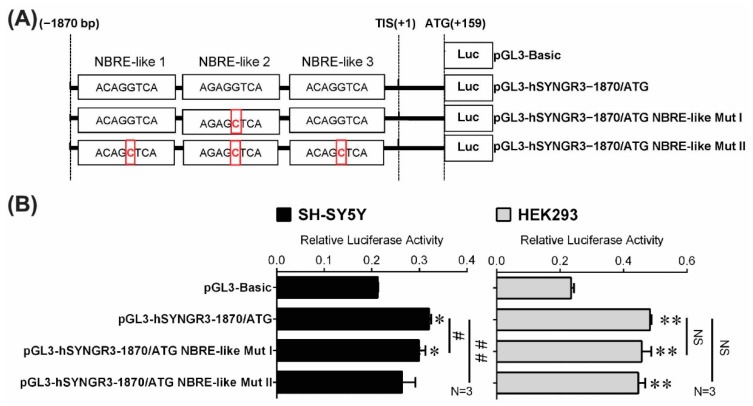
Effects of putative ncNBRE sites on the transcription of SYNGR3. (**A**) Site-directed mutagenesis of the three putative ncNBRE sites in pGL3-luciferase plasmids. (**B**) Comparison of the relative luciferase activities of wildtype and mutated putative ncNBRE sites in both neuronal SH-SY5Y and non-neuronal HEK293 cells. Data are expressed as mean ± SEM of three independent measurements. * *p* < 0.05 and ** *p* < 0.01 represent the statistical significance compared with pGL3-Basic plasmid. # *p* < 0.05 and ## *p* < 0.01 represent the statistical significance compared with native (pGL3-hSYNGR3–1870/ATG) construct. NS means no significance.

**Figure 6 ijms-23-03646-f006:**
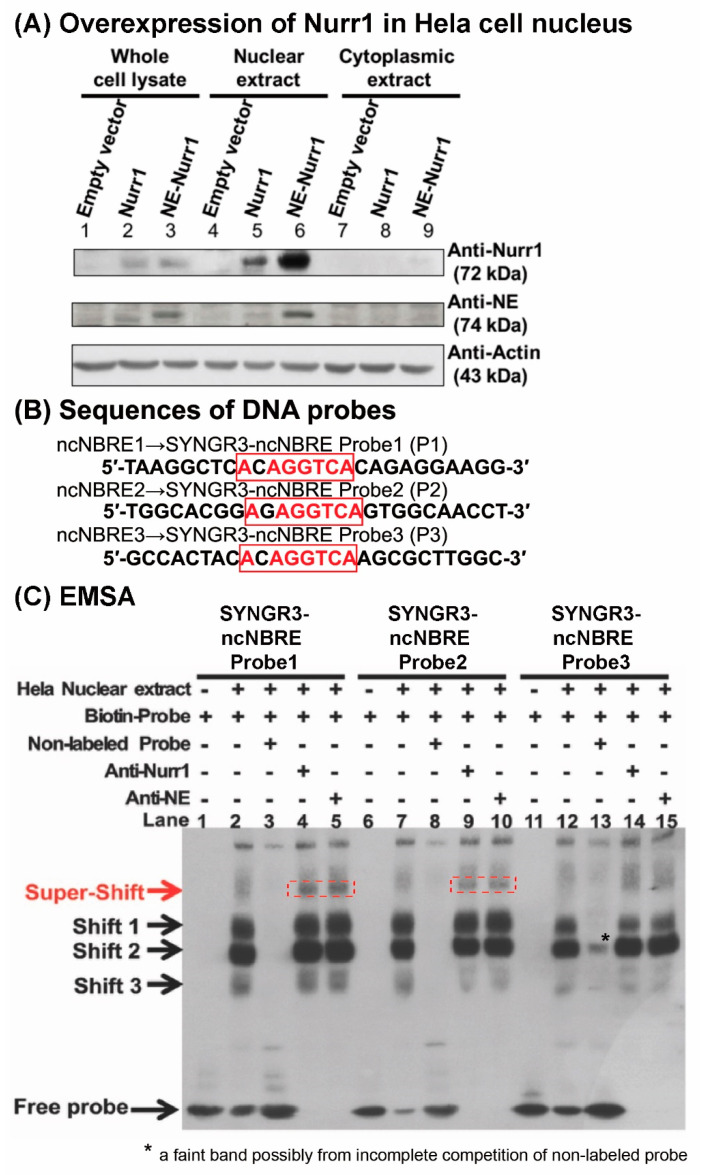
Probe design and gel shift assay of the three putative ncNBRE sites in the 5′-flanking region of human SYNGR3. (**A**) Western blot analysis of NURR1 and NE-NURR1 expression in the whole cell lysate (lane 1–3), nuclear extracts (lane 4–6), and cytoplasmic extracts (lane 7–9) from NURR1 and NE-tagged NURR1 overexpressing and empty vector control Hela cells. (**B**) sequences of three DNA probes, with putative ncNBRE sites (red). (**C**) EMSA shows specific protein–probe binding indicated by shifted bands 1–3. The specificity of NURR1 binding was confirmed by supershift bands with both anti-NURR1 and anti-NE antibodies (red dashed boxes). “NE” is a synthetic protein tag attached to an overexpressed NURR1 protein in a nuclear extract.

**Figure 7 ijms-23-03646-f007:**
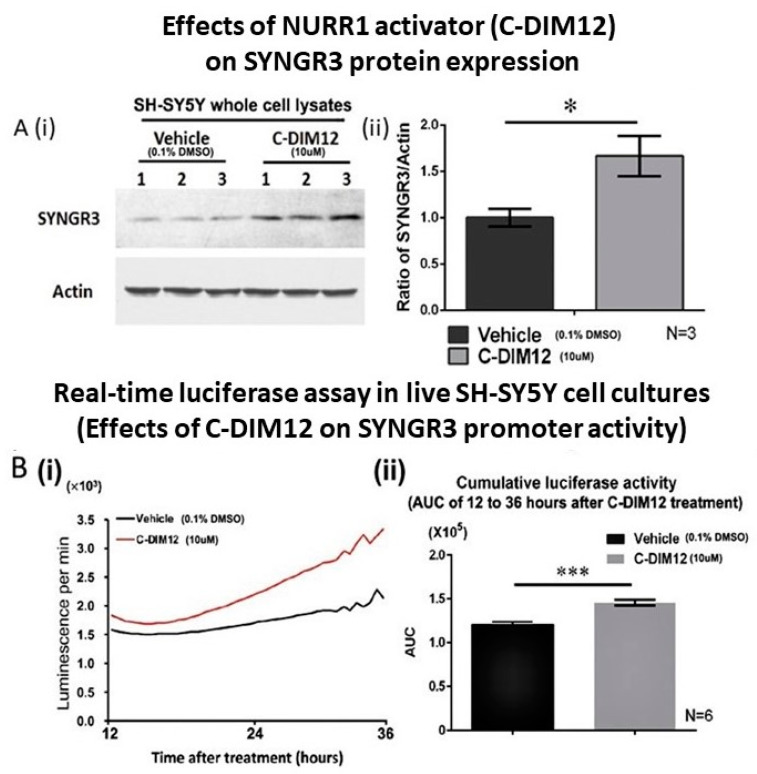
Effects of Nurr1 activator C-DIM12 on protein expression in SH-SY5Y cells and the effect of activator activity assayed in real time. (**A**) (i) total cell lysates from vehicle-treated and C-DIM12 treated SH-SY5Y cells were detected by anti-SYNGR3 and anti-Actin antibodies. The Western blots showed bands at 25 kDa corresponding to SYNGR3 and bands at 43 kDa corresponding to actin. SYNGR3 protein expression increased in SH-SY5Y cells treated with C-DIM12 compared with that in cells treated with vehicle. (ii) quantitative measurement of the SYNGR3 expression levels in C-DIM12 and vehicle-treated cells. (**B**) (i) schematic diagram of the luciferase reporter plasmids cloned with the 2029-bp 5′-flanking region of pGL3-hSYNGR3–1870/ATG containing three NBRE-like elements. SH-SY5Y cells were transiently transfected with the pGL3-hSYNGR3–1870/ATG construct containing three NBRE-like elements. Real-time luciferase activity was measured from 12 h to 36 h following treatment with vehicle and C-DIM12 (10μM) in SH-SY5Y cells. (ii) Cumulative luciferase activity was calculated from the area under the curve (AUC) of the real-time luciferase activity of pGL3-hSYNGR3–1870/ATG in SH-SY5Y cells treated with vehicle and C-DIM12. Values are mean ± SEM of six independent experiments. *** indicates significance at *p* < 0.01 as compared to vehicle-treated controls. * indicates significance at *p* < 0.05 compared with the level in vehicle-treated controls. The bars indicate mean ± SEM of the three independent experiments.

**Figure 8 ijms-23-03646-f008:**
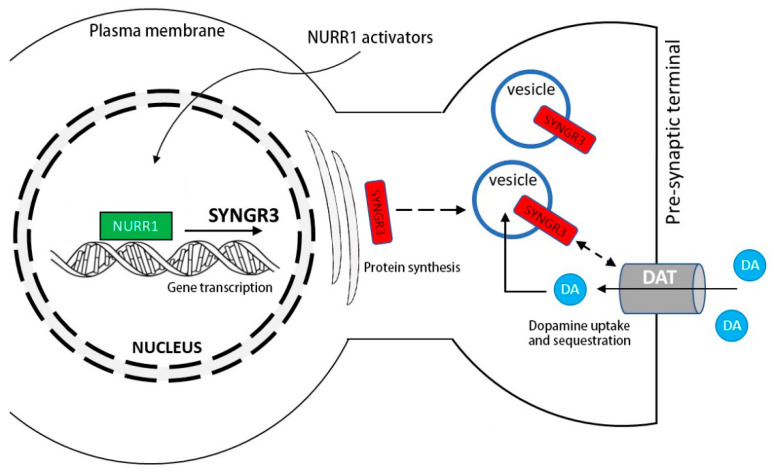
Hypothetic schematic diagram showing a potential role of NURR1 on SYNGR3 expression and dopamine (DA) homeostasis via its interaction with the dopamine transporter (DAT). NURR1 activators may be helpful to maintain dopaminergic synaptic function via SYNGR3, thus presenting a potential therapeutic target to preserve neuronal function in PD. DA: dopamine.

**Table 1 ijms-23-03646-t001:** List of genes showing the interaction and regulation of SYNGR3 (referenced from Pathways Common (https://www.pathwaycommons.org/), a public pathway and interaction database, accessed on 1 March 2022) [7]. The seven transacting factors which exert control of SYNGR3 transcription are highlighted.

Gene Name	Interact with SYNGR3	Regulate SYNGR3 Transcription
ACSF2	√	
ATF2		√
ATF3		√
ATF4		√
ATP1B4	√	
CREB1		√
E4F1		√
EHHADH	√	
ESR2	√	
GAD2	√	
GLP1R	√	
HSBP1L1	√	
IRF3	√	
JUN		√
MAPT	√	
MIDN	√	
MPP1	√	
NDRG4	√	
NOG		√
PLIN3	√	
PNKP	√	
SH3GLB1	√	
SLC39A9	√	
SPG21	√	
TTPA	√	

**Table 2 ijms-23-03646-t002:** Primers used for constructing SYNGR3 promoter–luciferase fusion plasmids.

Constructs	Primer	Sequence (5′→3′)
pCR4-TOPO-SYNGR3–1870/+224	Forward	TTTGAAGGCATCCAGGCCAGTGGAACTCTAGTGCAAGGAAAAGTT
	Reverse	CCGCGCAAAGCTCACGGGGTCCAG
pGL3-SYNGR3–1870/ATG	Forward	TTTGAAGGCATCCAGGCCAGTGGAACTCTAGTGCAAGGAAAAGTT
	Reverse	CATGCCATGGCCGGCCCGGGCGGGACG
pGL3-SYNGR3–1870/TIS	Forward	TTTGAAGGCATCCAGGCCAGTGGAACTCTAGTGCAAGGAAAAGTT
	Reverse	CCGCTCGAGCGGGGGACGCGCGGGACGGGGCG
pcDNA3.1(+)-NE-NURR1	Forward	CCCAAGCTTATGACCAAAGAAAACCCGCGTAGCAACCAGGAAGAAAGCTATGATGATAACGAAAGCCCTTGTGTTCAGGCGC
	Reverse	CGCGGATCCTTAGAAAGGTAAAGTGTCCAGG
pcDNA3.1(+)-NURR1	Forward	CCCAAGCTTATGCCTTGTGTTCAGGCGCAG
	Reverse	CGCGGATCCTTAGAAAGGTAAAGTGTCCAGG

**Table 3 ijms-23-03646-t003:** Primers used for the site-directed mutagenesis of pGL-SYNGR3 constructs.

Constructs	Target Site to Be Mutated	Primer	Sequence
pGL3-SYNGR3–XCPE1-Mut	XCPE1	Forward	GCGCGCGTCCAGCCCGGGCCG
		Reverse	CGGCCCGGGCTGGACGCGCGC
pGL3-hSYNGR3–1870/ATG ncNBRE Mut I	ncNBRE1	Forward	GATTGGCACGGAGAGCTCAGTGGCAACCTTG
		Reverse	CAAGGTTGCCACTGAGCTCTCCGTGCCAATC
pGL3-hSYNGR3–1870/ATG ncNBRE Mut II	ncNBRE2	Forward	TCCTAAGGCTCACAGCTCACAGAGGAAGGAC
		Reverse	GTCCTTCCTCTGTGAGCTGTGAGCCTTAGGA
	ncNBRE3	Forward	CGCCACTACACAGCTCAAGCGCTTGGC
		Reverse	GCCAAGCGCTTGAGCTGTGTAGTGGCG

## Data Availability

Not applicable.

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
