# Peer review of "Transcriptional Regulation of the Synaptic Vesicle Protein Synaptogyrin-3 (SYNGR3) Gene: The Effects of NURR1 on Its Expression"

_ijms, 2022, doi:10.3390/ijms23073646_

Round 1

Reviewer 1 Report

The manuscript is clearly written and the results crrespond to the claims presented.

I have some comments to the manuscript:

Table1:

Why tau protein (MAPT gene) is not included in Table 1, when reference 22 states that tau interacts with SYNGR3?

page6 line 202: says "SDS-10%PAGE", whereas page 5 line 189 it is "10% SDS-PAGE". It should be unified and explained that its a ratio between acrylamid, bisacrylamid

2.9: Authors should justify why they have used SEM and not SD for statistical evaluations

Line 42: What is the role of the unstructured cytoplasmic part of SYNGR3 (orange in alphafold prediction on uniprot)- could be discussed in introduction.

Line 200: The [rocess of cell lysate preparation is not described. Could be specified

Reviewer 2 Report

This study by Li et al. used various approaches to show that NURR1 binds to the promoter region of SYNGR3 gene and that activating NURR1 enhances the protein expression of SYNGR3.

On the whole, the study is done well. It would have been strengthened if ChIP assays were incorporated to show that the transcriptional control is also at the in vivo level. It would also be good to show the effects of mutating NURR1-binding sites on the protein expression of SYNGR3.

In addition, there are a few points that the authors should address:

  1. Fig. 5C EMSA experiment:
  • Lanes 2, 7, and 12 should not have any supershift bands. The presence of these bands makes their assignment of “super-shift” questionable. This needs to be explained.
  • On line 457, the authors claimed that “only one supershift band was seen”. Yet, lanes 4, 5, 9, 10, 14 and 15 all show more than one seemingly supershift bands, which are also faintly seen in lanes 2, 7, and 12, where there should be none. Thus, the statement of “only one supershift band” needs to be clarified.
  • In general, in supershift lanes, the shift bands should be lighter than those in lanes without antibodies. However, there are no detectable difference between the two in all 3 probes for shifts 1 and 2, and only a slight difference for shift 3 of Probe 2 and maybe probe 3. Densitometric measurements may help to clarify this.
  • Lanes 9, 10, 14, and 15 appear to have more than one “shift 3” band. Explanation is needed.
  • Lane 13 should not have any shift band, yet it clearly shows a band in the position of “Shift 2”.
  1. The uncertainties in the EMSA figure may be offset by an improved EMSA experiment and/or an in vivo ChIP assay using probes from the three putative NURR1-binding sites.
  2. It would be good to briefly discuss the possibility that NURR1 may interact with other transcription factor and/or coactivators.
  3. Lines 445-446: Explanation of NE should be in the Materials and Methods section.
  4. Typographical errors:
  • Line 448: Should be Fig. 5C, lanes 2, 7, and 12 (not Fig. 6C, lanes 2, 7 & 1).
  • Line 464: Should be Fig. 5 (not 6).
  • The term “supershift” is sometimes spelled with a hyphen and sometimes not. It should be consistent throughout the paper (including figure labels).

Reviewer 3 Report

My suggestions:

  1. Were there any disease-related mutations found in SYNGR3? I would mention them briefly in a paragraph or add a table on the mutations.
  2. I would also add a schematic figure of SYNGR3 and its domains and function. 
  3. In discussion, a schematic figure may be useful, which could explain the pathways, related to SYNGR3 defects (altered expression)
  4. Were there any mutations described in NURR1, related to PD or any neurodegenerative disease? You may mention them briefly in the discussion.

Round 2

Reviewer 3 Report

Authors fulfilled my suggestions. Thank you.